# Deep Learning Technique-Enabled Web Application Firewall for the Detection of Web Attacks [note 1]

**DOI:** 10.3390/s23042073

**Published:** 2023-02-12

**Authors:** Babu R. Dawadi, Bibek Adhikari, Devesh K. Srivastava

**Affiliations:** 1Department of Electronics and Computer Engineering, Pulchowk Campus, Tribhuvan University, Kathmandu 19758, Nepal; 2Department of Information Technology, Manipal University, Jaypur 303007, India

**Keywords:** WAF, LSTM, XSS, SQL injection, web security

## Abstract

New techniques and tactics are being used to gain unauthorized access to the web that harm, steal, and destroy information. Protecting the system from many threats such as DDoS, SQL injection, cross-site scripting, etc., is always a challenging issue. This research work makes a comparative analysis between normal HTTP traffic and attack traffic that identifies attack-indicating parameters and features. Different features of standard datasets ISCX, CISC, and CICDDoS were analyzed and attack and normal traffic were compared by taking different parameters into consideration. A layered architecture model for DDoS, XSS, and SQL injection attack detection was developed using a dataset collected from the simulation environment. In the long short-term memory (LSTM)-based layered architecture, the first layer was the DDoS detection model designed with an accuracy of 97.57% and the second was the XSS and SQL injection layer with an obtained accuracy of 89.34%. The higher rate of HTTP traffic was investigated first and filtered out, and then passed to the second layer. The web application firewall (WAF) adds an extra layer of security to the web application by providing application-level filtering that cannot be achieved by the traditional network firewall system.

## 1. Introduction

One of the common difficulties in various disciplines of computer science is protecting computers and networks from infiltration, theft, and disturbance [1]. The importance of a security system increases as the number of internet users increases. A web application firewall (WAF) acts as a barrier between a web application and the client on the internet when it is deployed in front of a web application [2,3]. A WAF is a type of reverse proxy that protects the web server from being exposed to the client by detecting anomalous traffic in the WAF, while a proxy server acts as an intermediary to protect a client machine’s identity. A WAF is controlled by a set of rules known as policies and a pre-trained module to predict new incoming requests. By filtering harmful communications, these policies try to guard against application vulnerabilities. The usefulness of a WAF is derived in part from the speed and ease with which policy modifications may be deployed, allowing for a faster reaction to various attack vectors [4,5]. Figure 1 depicts the fundamental structure of a WAF.

Many attempts have been made to build various security solutions, such as intrusion detection systems (IDS) and firewalls [6]. In most of these cases, network layer firewalls and IDS do not inspect HTTP packets in the application layer [2]. As a result, they are incapable of fully safeguarding web servers. Web applications, especially in the cloud, are one of the most appealing targets for attackers looking to break into an organization’s information infrastructure. Internal data leaks, financial losses, and website manipulation can all result from an organization’s failure to implement web security. A WAF is an application to identify and prevent many types of attacks, such as SQL injections, cross-site scripting (XSS), and dynamic denial of service (DDoS) [4,7]. WAF uses IDS methods in the application layer to secure web applications.

Most of the current WAFs work on signature-based systems. Traditional firewall systems are not meeting the needs of the modern environment to properly filter out attacks over any kind of network [5,8]. Attackers are smarter and they find adaptable techniques and tactics to breach the system. Hence, a signature-based protection system is not always a reasonable solution [9,10]. The signature-based system only works for known attacks and threats, but cannot work on zero-day attacks [7]. If we are providing a service through the web, the entire service and business process depends on this web service system, in which any kind of DDoS attack would directly hamper the service, the business, and the economy. A deep learning-based web application firewall is developed by training the system, such that it is capable of detecting new attack vectors, tactics, and behaviors. It is capable to protect web applications based on the effort we have made while training the module [11]. Correctly identifying the threat is a challenging task; if any threat is detected incorrectly, this may highly impact the business process resulting in organizational loss.

We introduce a layered architecture of WAF. Generally, in the WAF, features from the incoming traffic are extracted and tested for different types of threat detection modules/signatures. Based on the incoming traffic, a higher rate of detection is filtered in the first layer, and only the filtered traffic from the first layer is processed in the second layer. As the nature of the attack is different for different attacks, extracting the required parameter based on the nature of the attack and predicting new requests using a pre-trained model would increase the performance and accuracy of the WAF. The major contributions of this research are as follows:WAF layered architecture was proposed for DDoS, SQL injection, and XSS detection in the web-based service system.The proposed model’s performance was evaluated and achieved 97.57% accuracy with DDoS detection and 89.34% accuracy with XSS/SQL injection detection.

The rest of this paper is organized as follows: Section 2 presents a background study with related work on web application firewall implementation practices. Section 3 provides the methodology of the current research, including system architecture, dataset preparation, and approach for analysis, whereas Section 4 presents the results and analyses. Section 5 concludes the paper.

## 2. Background and Related Work

A web application firewall engine consists of two modules: (a) a configuration module (CM) and (b) packet analyzer module (PAM). When live packets are received from the internet, rule files filter them from CM and passes the traffic to PAM. PAM analyzes the packets and extracts the features from the packet. Using previously trained data, it tests and identifies the nature/character of that packet. Therefore, only analyzed and allowed packets are passed through the PAM to the web application server. WAF can be deployed as hardware devices on virtual appliances or software running on the same web server as the web application or through the cloud. It operates using a particular set of rules called policies [12]. In each of these deployment models, the WAF is always placed in front of the web application, intercepting all traffic between the application and the Internet. Thus, these policies determine the WAF firewalls that look for the traffic behavior and decides what action needs to be taken with vulnerabilities. The WAF will continue scanning web applications and receive GET and POST requests to identify and filter HTTP requests with malicious activity [5,13,14]. Furthermore, an intelligent WAF can even request to identify whether the participant is a human or a bot. When vulnerabilities are found in the application, the WAF immediately patches them to automatically block attackers and malicious actors, for example, bots and attacked IP addresses. WAFs are the first line of defense against complex attacks that threaten the integrity of any business. The most effective and efficient solutions provide the following WAF capabilities [15]:**Input protection** provides a comprehensive application filter that accepts only valid user inputs.**HTTP validation** detects HTTP vulnerabilities and prevents attacks by setting up the validation rules.**Policies tailored to widely used applications** are set up according to specific requirements and need. Thus, it protects applications from vulnerabilities and also provides real-time insights about the traffic.**Data leakage prevention** provides an alert and prevents any kind of unusual traffic or data leakage by identifying, filtering, and shielding the private data.**Automated attack blocking** provides automation tools for blocking attacks by denying malicious traffic from entering the network.

Web application security [16] is needed for securing information, clients, and organizations from information robbery, interference in commerce progression, or other destructive actions that come from cyber crime. Web application security and protection approaches endeavor to ensure the security of applications using measures such as WAFs, multi-factor confirmation for clients, utilized security, and approval of threats to preserve client states.

Every website on the Internet is vulnerable to cyber-attacks. The dangers range from human error to sophisticated cyber-attacks carried out by an organized group of criminals. The major incentive for cyber attackers, according to Verizon’s data breach investigations report, is financial [17]. Whether we run an E-commerce site or a small simple company website, we are at risk of being attacked. Each harmful assault on our website is unique, and with so many types of attacks circulating, it becomes difficult to defend against all of them. However, there is a lot that can be done to protect websites from these assaults and reduce the chances that dangerous hackers will target them. Major known web attacks are depicted in Figure 2. XSS and SQL injections are among the top web attacks.

DDoS attacks are deliberate attempts to interrupt the normal traffic of a targeted server, service, or network by flooding the target or its surrounding infrastructure with Internet traffic [18]. A DDoS attack is similar to unanticipated traffic congestion that prevents regular traffic from reaching its target. PCs and other networked resources, such as IoT devices, are used to flood the target with internet traffic that is controlled/instructed by the central system. Bots are individual devices, whereas a botnet is a collection of bots. Once a botnet has been formed, the attacker can lead an attack by sending remote commands to each bot. When a botnet targets a victim’s server or network, each bot sends requests to the target’s IP address, potentially overwhelming the server or network and preventing normal traffic from passing through. As each bot is a legal Internet device, it makes it difficult to distinguish attack traffic from typical Internet traffic. XSS is an injection attack that occurs when the attacker uses vulnerabilities in trusted websites to inject malicious code, and this code can be implemented to steal personal information from users, such as login information, session cookies, and sensitive information [1]. It can even remain on the website permanently to continue targeting multiple users. An SQL injection is a sort of online security issue that allows an attacker to manipulate database queries in a web application. It gives an attacker access to data they would not normally have access to. An attacker can change or erase this data in many cases, causing the application’s content or behavior to be permanently changed [3].

LSTM model is a special type of recurrent neural network. It is capable of learning long-term dependencies during the training of the module [7,19]. A LSTM model consists of three layers integrated together in each cell to process the input from sequential input data and output from the previous cell.$h_{t-1}$ = Previous cell output$x_{t}$ = Input in current cell$C_{t-1}$ = Previous cell state

The structure of LSTM network is depicted in Figure 3. The first layer is the forgotten gate. This layer filters out the content that needs to be memorized or not. The sigma function gives the output ($f_{t}$) ‘1’ or ‘0’. ‘1’ signals to memorize the previous cell, whereas ‘0’ signals to forget it.
(1)$$f_{t}=\sigma \left(W_{f}\cdot \left[h_{t-1},x_{t}\right]+b_{f}\right)$$

The second is the input gate that contains two layers, of which the first is the input gate sigmoid layer ($i_{t}$) that decides on which value to update and the second one is the “tanh” function. Both are combined and added to the previous layer output to give the cell state output ($\tilde{C_{t}}$).
(2)$$i_{t}=\sigma \left(W_{i}\cdot \left[h_{t-1},x_{t}\right]+b_{i}\right)$$
(3)$$\tilde{C_{t}}=tanh(W_{C}\cdot \left[h_{t-1},x_{t}\right]+b_{C}$$

The final layer is the output gate, which gives the output of the cell. The output of the sigmoid is obtained by the sigmoid output of the input and the previous cell’s output ($o_{t}$). The cell state value is passed through the “tanh” function. It multiplies with the sigmoid output ($o_{t}$) to get the cell output value ($h_{t}$).
(4)$$o_{t}=\sigma (W_{o}\left[h_{t-1},x_{t}\right]+b_{o}$$
(5)$$h_{t}=o_{t}\cdot tanh\left(\tilde{C_{t}}\right)$$

### Related Work

Gustavo et al. [20] explored the deep learning techniques implemented in web application firewalls to classify the HTTP traffic. The author used a transformer encoder to analyze the classification of HTTP traffic. Using natural language processing, the authors trained the model by transferring the HTTP traffic to the feature vector.

Moradi et al. [2] used a stacked auto-encoder method in the deep belief network to detect bad HTTP requests. The authors used the n-gram feature extraction model to extract features for model development. Three different machine learning models have been used with the CSIC 2010 and ECML/PKDD 2007 dataset, and compared the performance of these models to verify which had better performance as a web application firewall in the detection of anomalies.

Pen et al. [21] presented the importance of an unsupervised method of machine learning over a supervised learning method for attack detection. The authors proposed an auto-encoder-based model for the detection of such attacks to analyze XSS and SQL injections.

Rajesh et al. [22] analyzed different features including UDP flood attacks, ICMP ping flood attacks, TCP SYN flood attacks, and land attacks to distinguish between normal and DDoS attack traffic. The authors also presented a comparative analysis of the different machine learning methods, including K-nearest neighbour, decision tree, random forest, and naive Bayes.

Lente et al. [23] proposed a new model called 3C-LSTM, which is a combination of LSTM and CNN, claiming it had better accuracy than other models. The authors used the proposed model for XSS detection, trained by converting words to vectors. This work evaluated the model for different sizes of batch input, and proposed the best batch size for better results.

Keracan et al. [24] proposed using DA-SANA to detect attack traffic by considering the noise coefficient. The author used three datasets, CISC, PKDD, and a generated dataset to analyze the model to present the comparative results. In this work, the authors analyzed attacks including SQL injection, XSS, RCE, CSRF, XXE, and many more.

Liang et al. [9] worked on analyzing URL content and identifying whether a URL had an SQL injection and XSS payload or not. For this, they tokenized and vectorized the URL and used this information to train RNN, LSTM, and GRU machine learning modules.

Tekerek et al. [25] used the CSIC2010v2 dataset to train the CNN, and discussed the advantages of using CNN over ANN. The authors claimed that the proposed deep learning model had higher accuracy than other machine learning models.

To the best of our knowledge, there have been many studies carried out investigating TCP, UDP, SYN, and NTP flood types of DDoS attacks, but not specifically HTTP flood DDoS attacks. Hence, we investigated HTTP flood DDoS attacks and correlated two types of attacks, XSS and SQL injection, with one affecting the availability of service and the other affecting the confidentiality and integrity of the web services.

## 3. Methodology

### 3.1. System Working Architecture

A WAF exists between the web server and client. Incoming HTTP traffic is parsed and analyzed in the request processing unit of the WAF. The WAF was trained with a training dataset to predict whether new incoming HTTP traffic was normal or malicious. As the nature of a DDoS attack is different from XSS and SQL injections, the system was trained with a separate, appropriate dataset for these attacks. A new HTTP request is parsed and its parameters are extracted for prediction by the module. Then, it applies to the pre-trained module for a prediction. If the HTTP traffic is classified as malicious, then it will be dropped; otherwise, it will be passed to the second module for testing the SQL injection and XSS. Similarly, the second module identifies whether the HTTP traffic is normal or malicious. If the HTTP traffic is predicted to be normal traffic, the HTTP session is passed to the web server; otherwise, the HTTP session will be discarded/dropped in the WAF itself. As the rate of DDoS requests are very high if we check for DDoS in the first layer of the WAF, there is a higher rate of traffic filtered out in the first layer. This helps to increase the accuracy and performance of the WAF system. Our proposed WAF consists of two modules in the layered architecture, one for DDoS attack detection in the first layer and another for SQL injection and XSS detection in the second layer. Rather than training the module with the single dataset, training the module with separate datasets would lead to better results, as the nature of the data and attacks are different. The complete framework of this model is presented in Figure 4.

### 3.2. Framework of the Proposed Model

For the training phase, two different datasets were used. Different features and parameters were used for the detection of DDoS, XSS, and SQL injection, as per the nature of attack. One dataset contained DDoS detection parameters, such as time-to-live, packet length, request type, time, etc., whereas the SQL injection and XSS extracted the HTTP header and body parts to analyze the characteristics presented. Training and testing datasets were applied to the LSTM model using the generated dataset. After training the system, the weight of the node in the module was adjusted so that we could predict new requests by applying the pre-trained module. The module steps are shown in Figure 5.

**Decoder:** The captured data is in raw form, and needs to be decoded to a standard format. The DDoS data was decoded by Wireshark, whereas the SQL injection and XSS log were decoded using URL decoding.**Feature and parameters selection:** The decoded dataset consists of different features and parameters, so the appropriate parameters/features must be selected for training the module. The DDoS attack detection features were selected by analyzing standard dataset and correlation analyses in the captured data. For the SQL injection and XSS detection, we analyzed the standard dataset to perform a comparative analysis of normal and attack traffic, and selected the appropriate parameters.**Numericalization:** We coded the request methods of GET and POST as 1 and 2, respectively. Similarly, flag values in textual forms were transformed to 1 and 0, respectively.**Normalization/Scaling:** To reduce the complexity of the module, higher numerical values were normalized to lower ones using min-max normalization, which is $z_{i}$ = $\frac{x_{i}-min\left(x\right)}{max(x)-min(x)}$. Moreover, the data was converted to scalar form, which is suitable for the LSTM module to make sequential inputs.**LSTM module:** In DDoS attack detection, we need to use a large sequential dataset for the output and this sequence data are dependent on each other. Hence, instead of a normal feedforward network model, LSTM could be the better choice.

### 3.3. Data Collection Methodology

The standard datasets DDoS IDS ISCX 2012 and CIC-DDoS 2019 were used for the analysis and to detect the normal and DDoS attack traffic. Similarly, for the analysis of SQL injections and XSS detection, we used the CISC dataset as the standard dataset to identify features that could be indicators of XSS and SQL injection attacks. Additionally, we developed a simulation environment for DDoS, XSS, and SQL injection-related dataset generation for the processing and training of the WAF model.

#### 3.3.1. Dataset Preparation for SQL Injection and XSS

A test environment was established using the DVWA [26] web application vulnerability analysis tool, using the Burp Suite tool to pass the XSS and SQL injection payloads and capture the traffic in the middle proxy. The dataset generation steps are shown in Figure 6. Different XSS and SQL injection payloads were passed through the web browser. The traffic forwarded from the web browser to the server were captured in the middleware Burp Suite proxy. The captured raw traffic was then processed to extract the required parameters from the raw log. Then, we analyzed the captured traffic and extracted the features/parameters from it using around 5700 payloads to collect the HTTP attack traffic. The normal traffic was collected with normal input from the user interface.

#### 3.3.2. Data Collection for DDoS

The Low Orbit Ion Cannon (LOIC) tool and hulk tool in the Kali Linux environment in VMware was set up and implemented for data collection. We hosted a sample E-commerce site on a local host on a Windows system, then flooded the traffic from four LOIC instances consisting of two instances of LOIC on each machine. The forwarded traffic was captured with the Wireshark and then processed to extract useful information from it to train the WAF module.

#### 3.3.3. Correlative Data Collection for DDoS and SQL Injection

For the correlative analysis, the first layer of defense was for DDoS protection, whereas the second layer of protection was for XSS and SQL injection. SQL injection data acts like normal data in the DDoS attack layer of protection. Different payloads were passed from the browser as HTTP requests through the Burp Suite proxy and collected the HTTP traffic using Wireshark. The collected log in the respective tool was used for training the model. The collected data in Wireshark was taken as normal data, because it was collected by normally browsing the web. The correlated log was only applicable for the DDoS detection model, as the second layer of detection was applicable for XSS and SQL injection detection.

## 4. Results and Analysis

### 4.1. IDS ISCX 2012 Dataset

To identify the features that could be used to distinguish normal traffic from DDoS attack traffic, we used the ISCX dataset, taking a sample of 100 K normal data and 100 K DDoS attack data. From the samples, we analyzed the features of attack traffic and normal traffic. As shown in Figure 7, the first 100 K is the attack traffic and the last 100 K is the normal traffic. We analyzed 25 different features, from which the IP protocol used, time-to-live, Don’t Fragment flag, header length, source, and destination port used were the most distinguishable features between normal and DDoS attack traffic. As shown in Figure 7a, we found attack traffic with the UDP protocol used, whereas almost all normal traffic was with the TCP protocol. Similarly, we analyzed the TTL in the sample dataset. As shown in Figure 7b, we found that the normal dataset had a higher TTL value compared with the attack dataset. Similarly, during the analysis of header length, normal traffic had a higher header length compared with attack traffic, as shown in Figure 7c. In addition, we examined the Don’t Fragment flag used in forwarded packets in normal and attack traffic, as shown in Figure 7d. We found that attack traffic had a greater number of Don’t Fragment flags enabled compared with normal traffic. Moreover, there was greater variation in the destination ports used for the attack traffic compared with the normal traffic. For the other remaining features, we did not find distinguishing properties between normal and attack traffic.

### 4.2. 2019 DDoS CIC Dataset

Visualizations of the CIC dataset are presented in Figure 8a–d. We considered almost 100 K data samples from the CIC2019 dataset, where 80 K was the attack dataset and 20 K was the normal dataset. During the analysis of 45 normalized features, the most distinguishable parameters that differentiated between normal and attack traffic were push flag, flow rate, port used, protocol used, and urgent flag.

Referring to Figure 8a, it is obvious that the DDoS traffic flow rate is higher than for normal traffic, comparing the ratio between normal and attack traffic in the normalized data. As shown in Figure 8b, most of the normal traffic is forwarded with TCP, whereas attack traffic is forwarded with the UDP protocol. This is because TCP packets require an acknowledged response from the server, whereas UDP packets do not. Thus, the UDP is mostly used for attack traffic. In addition, we analyzed the total length of packet per second, which was far higher for attack traffic than for normal traffic, as shown in Figure 8c. Similarly, the most popular and lower-valued ports were used in the normal traffic, whereas there was a higher variation of ports used in attack traffic, as shown in Figure 8d. There was a variation in the segment length of normal traffic and DDoS attack traffic. We observed that small-length segments were bombarded at a higher rate with a DDoS attack. In addition, we observed more push and urgent flags were used in normal traffic compared with attack traffic.

### 4.3. Generated Dataset Representation

This dataset was collected in the simulation environment. A total of 100 K data samples were considered, out of which 50 K was the attack dataset and 50 K was the normal dataset. The flow rate sending the attack traffic from a single LOIC instance and the normal traffic flow from normal browsing are presented in Figure 9a,b. A total of 35 different features were extracted from the captured log and analyzed the feature patterns, as shown in Figure 10a–d. Other parameters considered to distinguish normal and attack traffic included flags, time-to-live, frame length, and packet length. These distinguishing features were used to train the model, so that it could be used for the prediction of new traffic.

### 4.4. XSS and SQL Injection Dataset

We analyzed the CISC 2010 normal and attack traffic dataset [27]. Normal HTTP traffic and HTTP traffic with XSS and SQL injection payloads were collected and the features were extracted by parsing the log collected in the proxy. To find appropriate parameters to distinguish between normal and attack traffic, we captured data with XSS and SQL injection payloads, normal browsing data, and data with normal input. We used a total of 4K XSS payloads and 2K SQL injection payloads for the generated attack dataset. The generated normal dataset was verified with the standard dataset by comparing the features and words presented in it. We considered the most frequent characteristics and words presented in the attack traffic with the payloads compared with the normal traffic.

### 4.5. Model with Generated Dataset

We used the generated dataset for the analysis, training, and testing of the module. We standardized the collected data using the standard score of a sample (x), which was calculated as: $z=\frac{\left(x-\eta \right)}{s},$ where ‘η’ is the mean and ‘s’ is the standard deviation. We made a matrix of m x n, where ‘m’ is the number of data presented and ‘n’ is the total number of selected features for model training, with a sliding window of size ‘z’ to separate continuous packets and reshape the data into a series of times. The data was reshaped into a three-dimensional matrix of shape (m-z) × z × n. This data was then implemented in the model with the train, test, and validation ratio of 60:20:20. The 60% dataset was used for the training of a bidirectional LSTM model with the activation function ‘tenh’. At the last layer of the LSTM model, we used a sigmoidal activation function for binary classification. For model regularization, we used the regularizer L2. In addition, we used the Adam optimizer for better optimization and used binary cross entropy as the loss function. As a result, after training the model up to 100 epochs, we observed an accuracy of 89.34%. The train and test accuracy of each epoch is shown in Figure 11.

Similarly, we applied the generated DDoS dataset to the LSTM model. After training the model up to 40 epochs, we observed an accuracy of 97.57%. The train and test accuracy of each epoch is shown in Figure 12.

### 4.6. Testing the Combined Model with New Test Dataset

For the validation of the model with the new dataset, we passed the log of live DDoS traffic to the new site and applied it to our pre-trained model for prediction. It was able to correctly detect 96% of the live traffic. Moreover, traffic with XSS and SQL injection was applied to the DDoS module, as it detected it as good traffic and passed it to the second layer. The features for prediction in the second layer were extracted. It correctly detected 90% of HTTP packets with the payloads. A proxy was configured on the Jupiter notebook and listened to the specific port for live HTTP traffic input to the system. When traffic was routed from source to destination, it passed through the two modules: first, it was checked for DDoS attacks, then for XSS and SQL injection attacks. Only correctly classified traffic was routed to the web server.

A confusion matrix of 2279 test datasets is shown in Figure 13. A total of 922 attacks were correctly predicted as malicious, whereas 1129 normal datasets were correctly predicted as normal. Similarly, seven normal datasets were detected as malicious by the model, and 221 malicious datasets were classified as normal.

A confusion matrix is shown in Figure 14 for about 53,133 DDoS datasets showing that 26,505 attacks were correctly classified as malicious, whereas 25,131 normal dataset were correctly classified as normal. Similarly, 1386 normal datasets were detected as malicious by the model, and 133 malicious dataset were classified as normal.

Comparing the performance between the first and second layer of detection, as shown in Figure 15, we found that the DDoS module had higher accuracy and recall than the XSS and SQL injection module, whereas the XSS and SQL injection module had higher precision than the DDoS detection module. The performance parameters of previous studies are shown in Table 1. Not one study considered the combination of all three attacks (XSS, SQL, and DDoS), which is what we addressed in our work. The performance of our proposed model is satisfactory compared with previous methods because of the correlative dataset obtained for analysis.

## 5. Conclusions

Using LSTM as our deep learning approach, the proposed model detected DDoS, XSS, and SQL injection attacks with considerably good accuracy. The first detection layer was a DDoS attack detection model with an accuracy of 97.57%, and the second layer was for XSS and SQL injection attack detection with an accuracy of 89.34%. We analyzed features and parameters for attack detection, which reduced false positives during traffic filtering in the WAF. As DDoS traffic comes at a higher rate than normal traffic, the system’s performance imporves when we check the traffic in a layered format, i.e., first checking for DDoS before testing for SQL injection and XSS. Moreover, we analyzed the performance perspective of the web application when an extra layer of filtering was added and found a slight impact on performance. However, this difference was not distinguishable from a user experience perspective.

This study focused on three types of web attacks: DDoS, SQL injection, and XSS. Future studies could include other types of common web attacks, such as RCE, malware, brute force, etc. Due to the similar detection properties, we examined SQL injection and XSS in a single module in this work. Further testing of models with other types of deep learning algorithms could lead to greater enhancements in WAF performance.

## Figures and Tables

**Figure 1 sensors-23-02073-f001:**
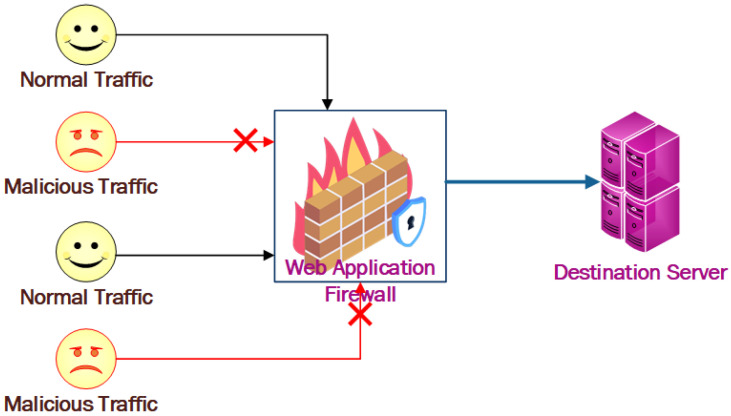
Working of web application firewall.

**Figure 2 sensors-23-02073-f002:**
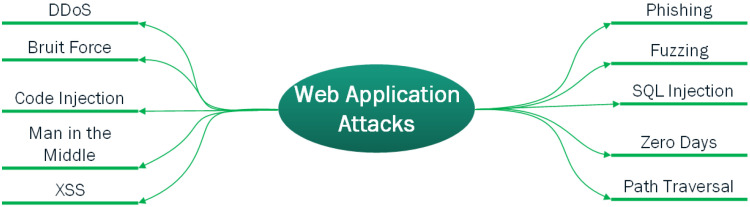
Different types of web attacks.

**Figure 3 sensors-23-02073-f003:**
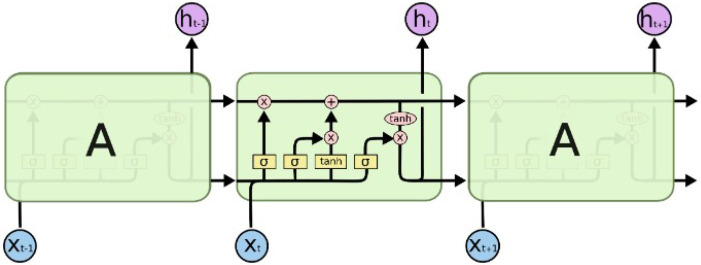
LSTM network.

**Figure 4 sensors-23-02073-f004:**
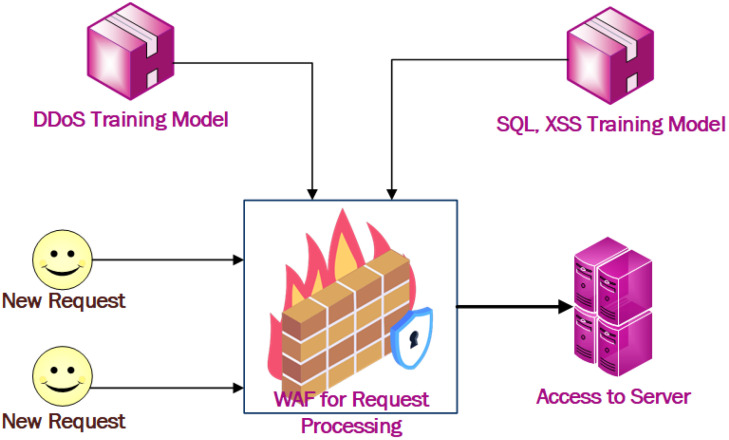
Model development framework.

**Figure 5 sensors-23-02073-f005:**
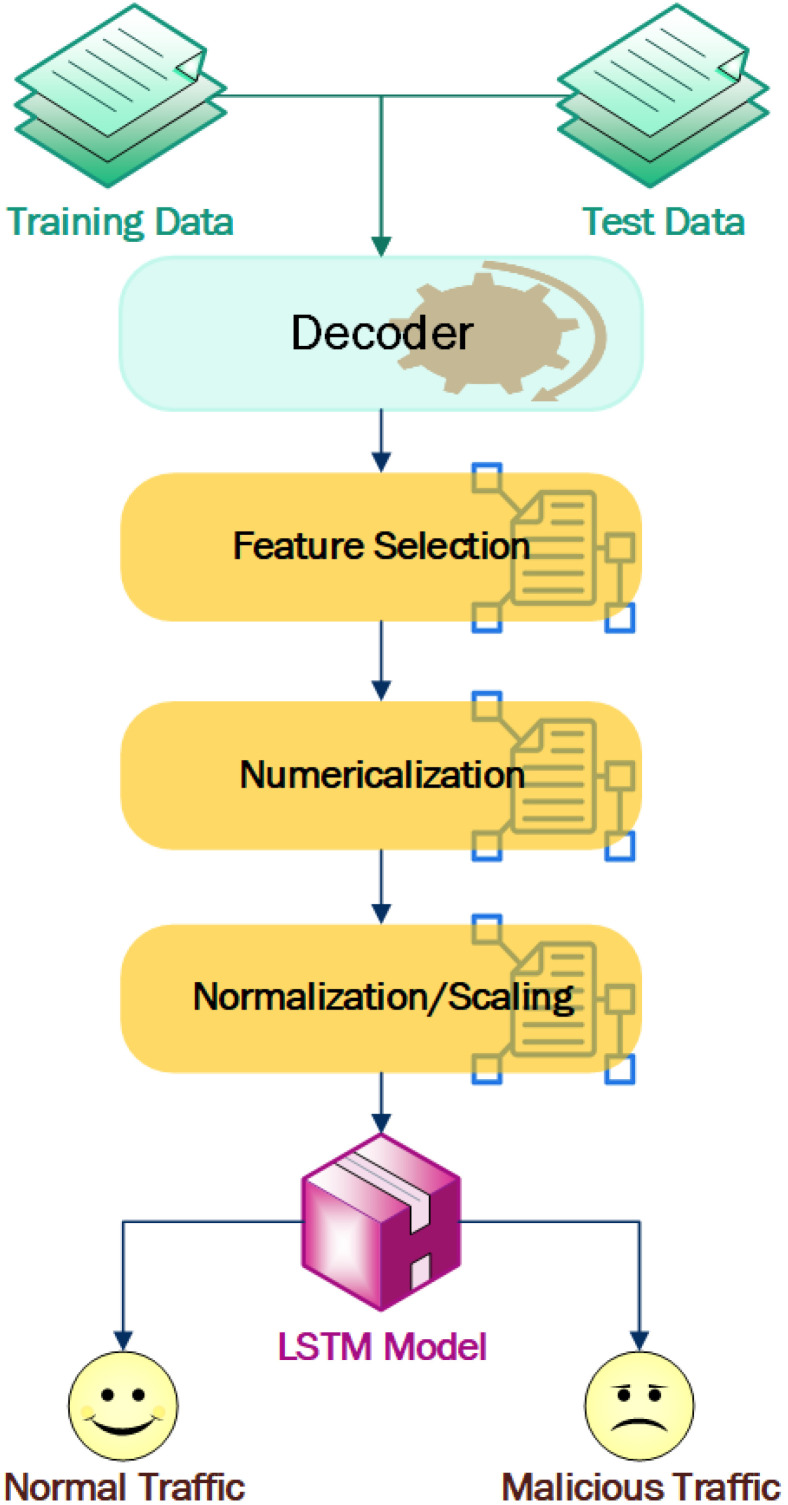
Flowchart of LSTM modules.

**Figure 6 sensors-23-02073-f006:**
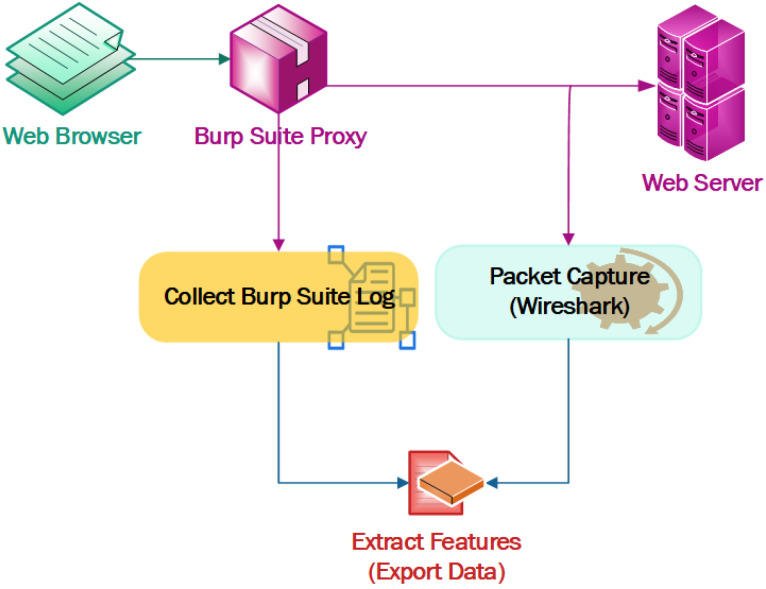
Dataset generation approach for DDoS, XSS, and SQL injection.

**Figure 7 sensors-23-02073-f007:**
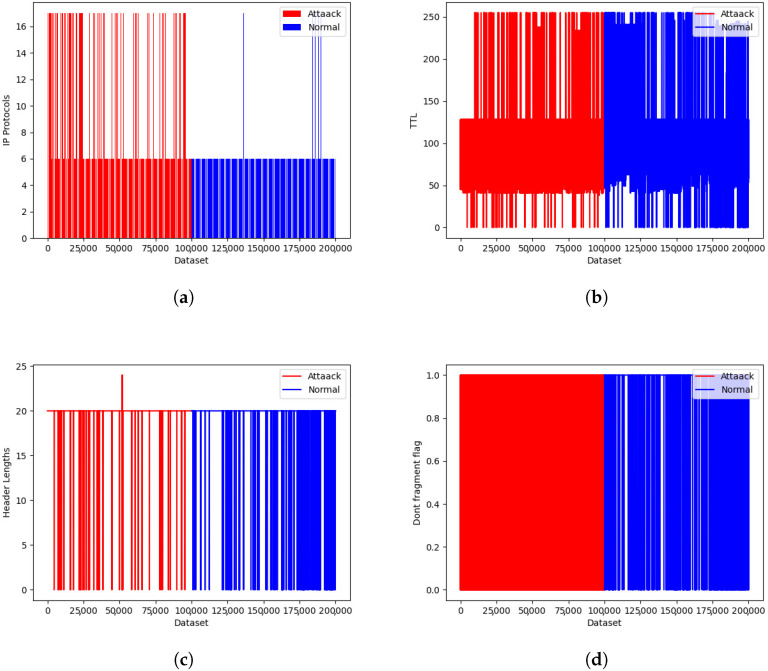
IDS ISCX 2012 dataset representations: (**a**) protocol used; (**b**) TTL value used; (**c**) header length; and (**d**) Don’t Fragment flag.

**Figure 8 sensors-23-02073-f008:**
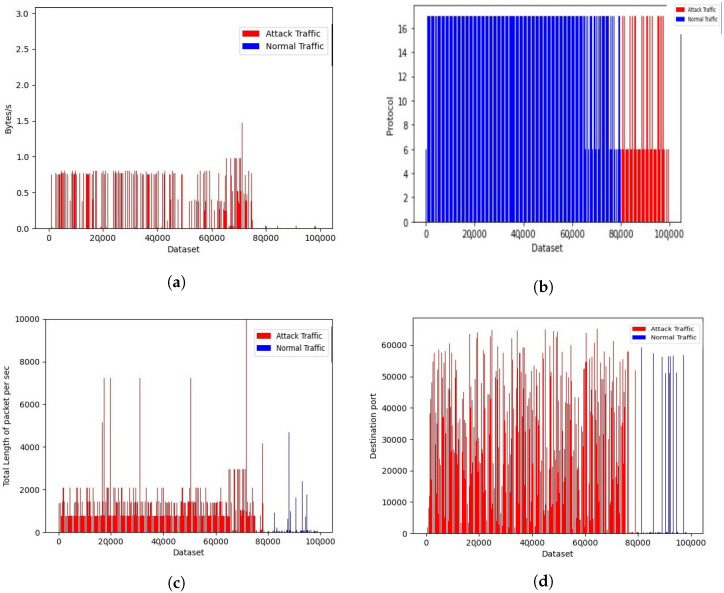
DDoS CIC 2019 dataset representation: (**a**) flow rate in normal vs. attack traffic; (**b**) protocol used; (**c**) total length of packet per sec; and (**d**) destination port used.

**Figure 9 sensors-23-02073-f009:**
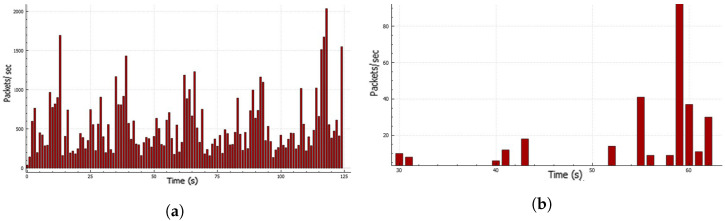
Rate representation during data preparation: (**a**) traffic rate from single LOIC instance and (**b**) rate during normal browsing of the site.

**Figure 10 sensors-23-02073-f010:**
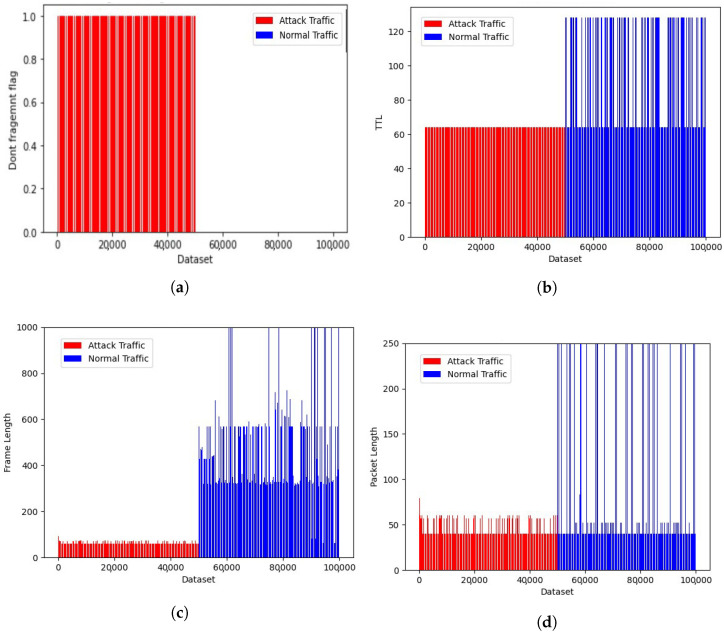
Dataset generated in the simulation environment representation: (**a**) Don’t Fragment flag; (**b**) time-to-live; (**c**) frame length; and (**d**) packet length.

**Figure 11 sensors-23-02073-f011:**
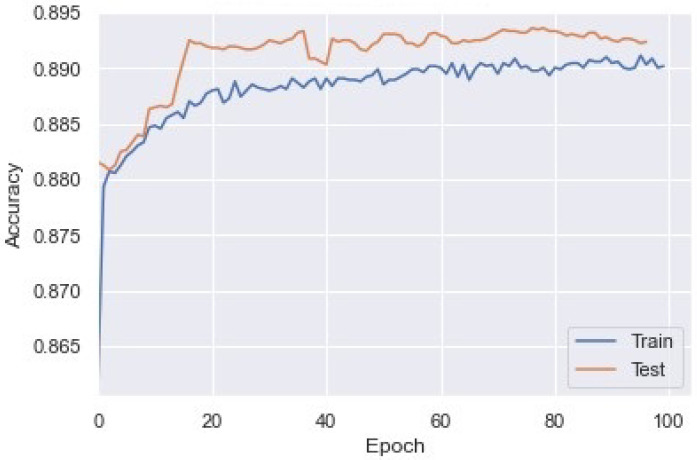
Training and test accuracy of LSTM model for XSS and SQL injection detection.

**Figure 12 sensors-23-02073-f012:**
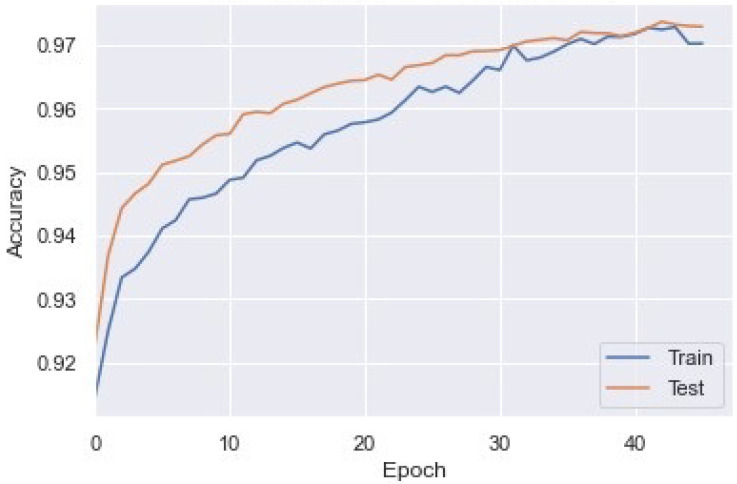
Training and test accuracy of LSTM model for DDoS detection.

**Figure 13 sensors-23-02073-f013:**
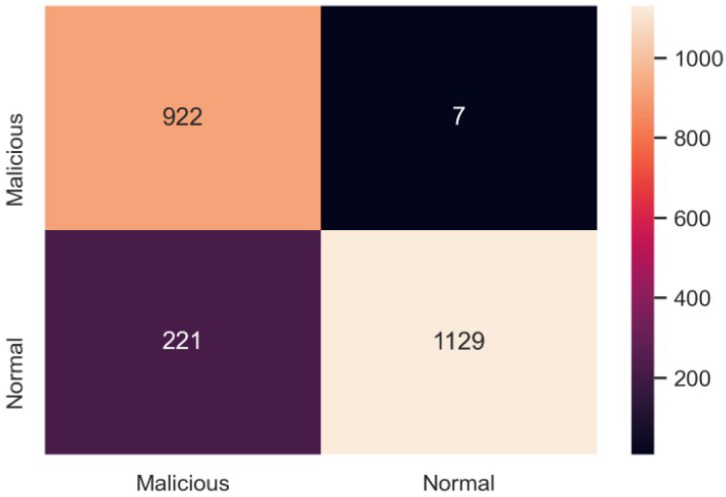
Confusion matrix for XSS and SQL injection detection model tested.

**Figure 14 sensors-23-02073-f014:**
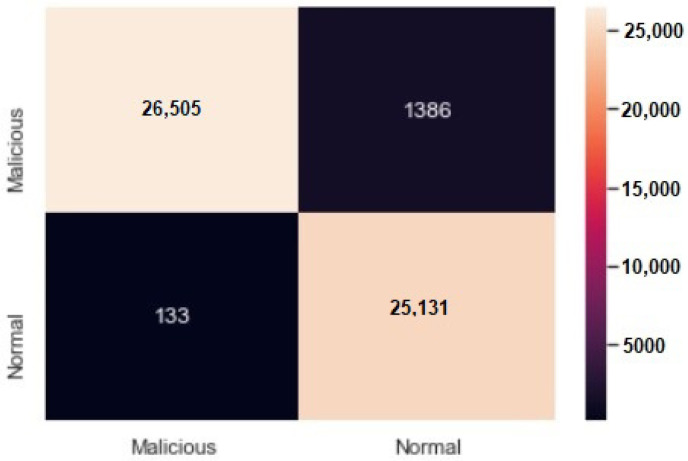
Confusion matrix for DDoS detection model tested.

**Figure 15 sensors-23-02073-f015:**
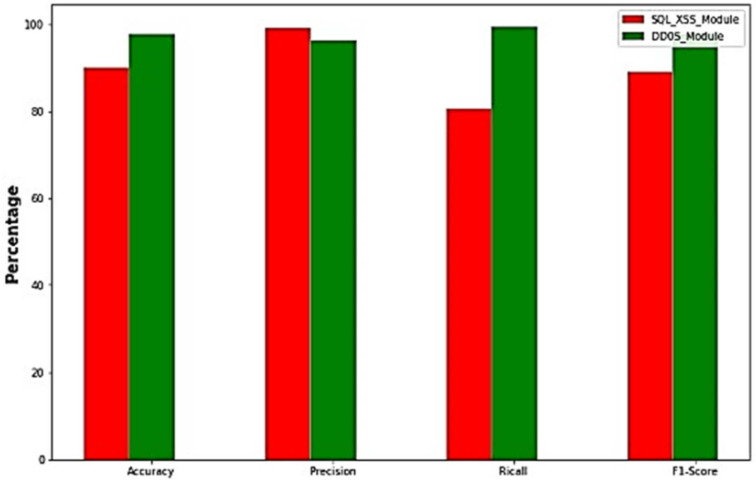
Performance evaluation of modules.

**Table 1 sensors-23-02073-t001:** Model performance comparison in terms of accuracy.

Ref.	XSS	XSS and SQL Injection	DDoS
[1]	99.5%	-	-
[7]	-	87.26%	-
[10]	-	-	99.22%
[23]	99.4%	-	-
[18]	-	-	97.36%
**Proposed**	-	**89.34%**	**97.57%**

## Data Availability

The program code, datasets, and experimental work snapshots of this study are available at our GitHub link: https://github.com/baburd/WAF.

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
