# Peer review of "Deep Learning Technique-Enabled Web Application Firewall for the Detection of Web Attacks†"

_sensors, 2023, doi:10.3390/s23042073_

Round 1

Reviewer 1 Report

The authors proposed a LSTM-based WAF for web attack prevention. They also conduct some experimental results to justify their claims. This reviewer thinks that the contribution of the proposed work is limited and/or not being clearly justified. Here are some comments:

-There are plethora of work using LSTM-based methods for web attack prevention such as:

A.M Vartouni et al., "Auto-Encoder LSTM Methods for AnomalyBased Web Application Firewall", IJICTR,2019.

I Kotenko et al., "LSTM Neural Networks for Detecting Anomalies Caused by Web Application Cyber Attacks", New Trends in Intelligent Software Methodologies, Tools and Techniques, 2021.

M. Krishnan et al.,"Detection and defending the XSS attack using novel hybrid stacking ensemble learning-based DNN approach",Digital Communications and Networks, 2022.

S. Hao, et al.,"BL-IDS: Detecting Web Attacks Using Bi-LSTM Model Based on Deep Learning", Springer series.

K. O. A. Alimi, et al.,"Refined LSTM Based Intrusion Detection for Denial-of-Service Attack in Internet of Things",J. Sens. Actuator Netw. 2022.

the authors has not cited and/or compared with these similar approaches.

-The quality of must figures is low and must be enhanced.

-English must be re-checked for typos.

Author Response

Respected Reviewer,
Thank you for your time and careful reviews. The review response is attached as PDF file.

Regard

Author(s)

Reviewer 2 Report

1. Some screenshot figures must be removed. The other figures are required to improve to show much more technique information.

2. The English representation is poor, such as "investigated first", "new request", "good request", "bad request". Polish the whole paper carefully.

3. What are the "first layer" and "second layer"? Why they work? In the method/experiment section, add more texts for them.

4. The introduction, relate works section should be improved to declare the motivations clearly.

Author Response

(The authors gave the same response as above.)

Reviewer 3 Report

This paper is motivated by the grows of cyberthreats in the emerging computing technologies. The authors claim proposing a WAF layered architecture to detect DDoS, SQL injection and XSS attacks in the web application. After reading this paper, I have the following comments:

Major Comments:

-       A motivational scenario is needed in the Introduction section. You must give strong motivation for providing your web application firewall architecture.

-       It is not mentioned in the paper, how do you decided/chosen the “Protocol used, Destination port used, Flow rate, Flow bytes, Push flag used, and Average segment size” as the most distinguishable features?

-       Section 2: Literature review is redundant; it should be deleted.

-       The second contribution mentioned in line: 62 must be rephrased after comparing your introduced architecture with similar related contributions.

-       The implementation details for the introduced WAF is missing.

-       The results have not been compared with other published contributions solving the same problem.

-       The methodology section contains some subsections with simple details that are unable to reflect the contribution of the paper mentioned in the abstract.

-       The methodology shown in this paper is just using some simple tools and showing their results.

-       There is no significant contribution showing the integration of the LSTM model in the WAF.

-       Extra experimental results are needed.

-       There are many missing references that must be cited in your paper and your results must be compared with their results.

-       The code used to build such architecture should be available on GitHub and its link should be shown elsewhere in this paper.

Minor Comments:

-       There are some typos and grammatical issues.

-       The figures quality is poor.

-    The references are not up to date.

Author Response

(The authors gave the same response as above.)

Round 2

Reviewer 1 Report

The authors have improved the paper.

Author Response

Respected reviewer,

Thank you very much!

Regards

Authors

Reviewer 2 Report

"SQL injection" of column 1 in Table 1 can be removed.

Author Response

Respected Reviewer,

"SQL injection" of column 1 in Table 1 has been removed. Thank you for pointing out situation and suggestions.

Regards

Authors

Reviewer 3 Report

This version is better than the original version. You should mention a more specific answer to the reviewer comments in the future. It is not good to find in most of the responses general words such as "We have massively revised the manuscript".

Again, all the figures must redrawn using a drawing software MS Visio or Drawio.

Also, for the results' figures, it must not be screenshots. You can use OriginLab or Matlab to redraw them.

Author Response

Respected Reviewer,

We apologize for the repetition of the terms/sentences. Thank you for your valuable comments and suggestions. The responses are attached in the separate PDF file.
